# INEQUALITY PHENOMENON IN $l_\infty$-ADVERSARIAL TRAINING, AND ITS UNREALIZED THREATS

Ranjie Duan[1], Yuefeng Chen[1], Yao Zhu[1,2], Xiaojun Jia[1,3], Rong Zhang[1] & Hue Xue[1]

[1]Alibaba Group, [2]Zhejiang University, [3] University of Chinese Academy of Sciences
{ranjie.drj, yuefeng.chenyf, stone.zhangr,
hui.xueh}@alibaba-inc.com
ee_zhuy@zju.edu.cn, jiaxiaojun@iie.ac.cn

## ABSTRACT

The appearance of adversarial examples raises attention from both academia and industry. Along with the attack-defense arms race, adversarial training is the most effective against adversarial examples. However, we find inequality phenomena occur during the $l_\infty$-adversarial training, that few features dominate the prediction made by the adversarially trained model. We systematically evaluate such inequality phenomena by extensive experiments and find such phenomena become more obvious when performing adversarial training with increasing adversarial strength (evaluated by $\epsilon$). We hypothesize such inequality phenomena make $l_\infty$-adversarially trained model less reliable than the standard trained model when the few important features are influenced. To validate our hypothesis, we proposed two simple attacks that either perturb important features with noise or occlusion. Experiments show that $l_\infty$-adversarially trained model can be easily attacked when a few important features are influenced. Our work sheds light on the limitation of the practicality of $l_\infty$-adversarial training.

## 1 INTRODUCTION

Szegedy et al. (2013) discovered adversarial examples of deep neural networks (DNNs), which pose significant threats to deep learning-based applications such as autonomous driving and face recognition. Prior to deploying DNN-based applications in real-world scenarios safely and securely, we must defend against adversarial examples. After the emergence of adversarial examples, several defensive strategies have been proposed (Guo et al., 2018; Prakash et al., 2018; Mummadi et al., 2019; Akhtar et al., 2018). By retraining adversarial samples generated in each training loop, adversarial training (Goodfellow et al., 2015; Zhang et al., 2019; Madry et al., 2018b) is regarded as the most effective defense against adversarial attacks. The most prevalent adversarial training is $l_\infty$ adversarial training, which applies adversarial samples with $l_\infty$ bounded perturbation by $\epsilon$.

Numerous works have been devoted to theoretical and empirical comprehension of adversarial training (Andriushchenko & Flammarion, 2020; Allen-Zhu & Li, 2022; Kim et al., 2021). For example, Ilyas et al. (2019) proposed that an adversarially trained model (robust model for short) learns robust features from adversarial examples and discards non-robust ones. Engstrom et al. (2019) also proposed that adversarial training forces the model learning to be invariant to features to which humans are also invariant. Therefore, adversarial training results in robust models' feature representations that are more comparable to humans. Theoretically validated by Chalasani et al. (2020), the $l_\infty$-adversarial training suppresses the significance of the redundant features, and the robust model, therefore, has sparser and better-behaved feature representations than the standard trained model. In general, previous research indicates that robust models have a sparse representation of features and view such sparse representation as advantageous because it is more human-aligned. Several works investigate this property of robust models and attempt to transfer such feature representation to a standard trained model using various methods (Ross & Doshi-Velez, 2018; Salman et al., 2020; Deng et al., 2021).

However, contrary to the claim of previous work regarding such sparse feature representation as an advantage, we find that such sparseness also indicates inequality phenomena (see Section 3.1 for detailed explanation) that may pose unanticipated threats to $l_\infty$-robust models. During $l_\infty$-adversarial training, the model not only suppresses the redundant features (Chalasani et al., 2020) but also suppresses the importance of other features including robust ones. The degree of suppression is proportional to the adversarial attack budget (evaluated by $\epsilon$). Hence, given the input images for an $l_\infty$-robust model, only a handful of features dominate the prediction. Intuitively, standard-trained models make decisions based on various features, and some redundant features serve as a "bulwark" when a few crucial features are accidentally distorted. However, in the case of a $l_\infty$ robust model, the decision is primarily determined by a small number of characteristics, so the prediction is susceptible to change when these significant characteristics are modified (see Figure 1). As shown in Figure 1, an $l_\infty$-robust model recognizes a street sign using very few regions of the sign. Even with very small occlusions, the robust model cannot recognize a street sign if we obscure the region that the model considers to be the most important (but well recognized by humans and the standard-trained model). Even if an autonomous vehicle is deployed with a robust model that achieves high adversarial robustness against worst-case adversarial examples, it will still be susceptible to small occlusions. Thus, the applicability of such a robust model is debatable.

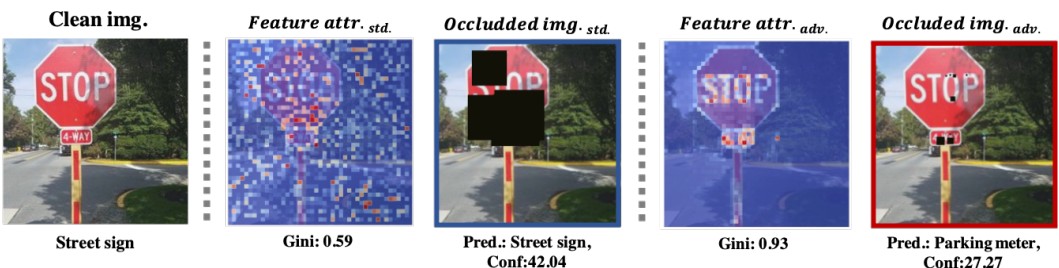

Figure 1: $l_\infty$-**robust model fails to recognize street sign with small occlusions**. With given feature attribution maps that attribute the importance of each pixel, we occlude the image's pixels of high importance with small patches. The resultant image fools the robust model successfully. We notice prior works (Tsipras et al.) showed that feature attribution maps of robust models are perceptually aligned. For clarity we strongly suggest the readers check Appendix A.2. )

In this work, we name such a phenomenon that only a few features are extremely crucial for models' recognition as "inequality phenomenon". we study the inequality from two aspects: 1) global inequality: characterized by the dominance of a small number of pixels. 2) regional inequality: characterized by the tendency of pixels deemed significant by the model to cluster in particular regions. We analyze such phenomena on ImageNet- and CIFAR10-trained models with various architectures. We further devise attacks to expose the vulnerabilities resulting from such inequality based on our findings. Experiments demonstrate that under the premise that human observers can recognize the resulting images, $l_\infty$-robust models are significantly more susceptible than the standard-trained models. Specifically, they are susceptible to occlusion and noise with error rates of 100% and 94% respectively, whereas standard-trained models are only affected by 30.1% and 34.5%. In summary, our contribution can be summed up in the following manner:

- We identify the occurrence of the inequality phenomenon during $l_\infty$-adversarial training. We design correlative indices and assess such inequality phenomena from various perspectives (global and regional). We systematically evaluate such phenomena by conducting extensive experiments on broad datasets and models.

- Then, we identify unrealized threats posed by such inequality phenomena that $l_\infty$-robust models are much more vulnerable than standard trained ones under inductive noise or occlusion. In this case, during the $l_\infty$-adversarial training, the adversarial robustness is achieved at the expense of another more practical robustness.

- Our work provides an intuitive understanding of the weakness of $l_\infty$-robust model's feature representation from a novel perspective. Moreover, our work sheds light on the limitation and the hardness of $l_\infty$-adversarial training.

## 2 BACKGROUND AND RELATED WORK

### 2.1 ADVERSARIAL ATTACK

DNNs are known to have various risks. These risks include adversarial attacks Li et al. (2021); Zhu et al. (2022); Mao et al. (2021); Qi et al.; Gu et al. (2022), backdoor attacks (Guo et al., 2023; Qi et al., 2023), privacy concerns (Li et al., 2020) and etc. Given a model denoted as $f(x;\theta) : x \to \mathbb{R}^k$ and training dataset denoted as $D$, empirical risk minimization (ERM) is a standard way (denoted as standard training) to train the model $f$ through:

$$\min_\theta E_{(x,y)\in D} loss(x,y) \qquad (1)$$

where $y \in \mathbb{R}^k$ is the one-hot label for the image and $loss\,(x,y)$ is usually cross-entropy loss. With such a training scheme, the model typically performs well on clean test samples. Adversarial examples (Szegedy et al., 2013) aim to generate perturbation superimposed on clean sample $x$ to fool a well-trained model $f$. Adversarial example $x'$ can be crafted by either following the direction of adversarial gradients (Goodfellow et al., 2015; Kurakin et al., 2016; Madry et al., 2018a; Duan et al., 2021) or optimizing perturbation with a given loss (Carlini & Wagner, 2017; Chen et al., 2018).

### 2.2 ADVERSARIAL TRAINING

Several defensive strategies are proposed to improve the models' adversarial robustness (Wong & Kolter, 2018; Akhtar et al., 2018; Meng & Chen, 2017; Raghunathan et al., 2018; Wu et al., 2022). However, analysis by Athalye et al. (2018) shows that among various defensive strategies against adversarial examples, only the adversarial training does not rely on the obfuscated gradient and truly increases the model's robustness. A model is considered robust against adversarial examples if:

$$argmax f\,(x;\theta) = argmax f(x+\sigma;\boldsymbol{\theta}), s.t. ||\sigma||_\infty \leq \epsilon \qquad (2)$$

where $\epsilon$ represents the magnitude of the perturbation. Therefore, the core idea of adversarial training is to train models with adversarial examples, formally:

$$loss(x,y) = E_{(x,y)\in D}[\max_{||\sigma||_\infty \leq \epsilon} loss(x+\sigma,y)], \qquad (3)$$

The objective $\max_{|\sigma|\leq\epsilon} loss(x+\sigma, y)$ introduces the model to minimize empirical risk on the training data points while also being locally stable in the (radius-$\epsilon$) neighborhood around each of data points $x$. The objective is approximated via gradient-based optimization methods such as PGD (Madry et al., 2018b). Several following works attempt to improve adversarial training from various aspects (Shafahi et al., 2019; Sriramanan et al., 2021; Jia et al., 2022b; Cui et al., 2021; Jia et al., 2022a;c).

Interestingly, Ilyas et al. (2019) proposes that by suppressing the importance of non-robust features, adversarial training makes the trained model more focused on robust and perceptually-aligned feature representations. In this process, the feature representation becomes more sparse. Chalasani et al. (2020); Salman et al. (2020); Utrera et al. (2020) suggests that the feature representation generated by a robust model is concise as it is sparse and human-friendly. It only assigns the feature that is truly predictive of the output with significant contributions.

### 2.3 HOW INEQUALITY FORMS DURING $l_\infty$ ADVERSARIAL TRAINING

In (Chalasani et al., 2020), they theoretically prove the connection between adversarial robustness and sparseness: During $l_\infty$-adversarial training, supposed the adversarial perturbation $\sigma$ satisfying $||\sigma||_\infty \leq \epsilon$, the model attempts to find robust features serving as strong signals against perturbation. Meanwhile, the non-robust ones which serve as relatively weak signals, and their significance (acquired by feature attribution methods) more aggressively shrunk toward zero. The shrinkage rate is proportional to adversaries' strength (evaluated by $\epsilon$). In other words, standard training can result in models where many non-robust features have significant importance for models, whereas $l_\infty$-adversarial training tends to selectively reduce the magnitude of the significance of non-robust features with weakly relevant or irrelevant signals and push their significance close to zero. In the end, the feature attribution maps generated by gradients-based feature attribution methods (Smilkov et al., 2017; Lundberg & Lee, 2017; Sundararajan et al., 2017) look more sparse. They regard such sparseness as a merit of adversarial training as it produces more concise and human-aligned feature attributions. However, we further study such sparseness and find it introduces a phenomenon of extreme inequality, which results in unanticipated threats to $l_\infty$-robust models.

## 3 METHODOLOGY

In this section, we first introduced the index used to measure inequality from two aspects. Then we propose two types of attacks to validate our hypothesis: extreme inequality brings in unexpected threats to $l_\infty$-robust model.

### 3.1 MEASURING THE INEQUALITY OF A TEST DATA POINT

Firstly, feature attribution maps are required to characterize the inequality degree by given test data point $x$ and model $f$. Several feature attribution methods have been proposed in recent years (Smilkov et al., 2017; Lundberg & Lee, 2017; Sundararajan et al., 2017). In general, feature attribution methods rank the input features according to their purported importance in model prediction. To be specific, we treat the input image $x$ as a set of pixels $x = \{x_i, i = 1...M\}$ and denote the generated feature attribution map of $x$ of model $f$ as $A^f(x)$, where $A^f(x)$ is composed of $a_i$. Feature attribution methods attribute an effect $a_i$ to each $x_i$, and summing the effects of all feature attributions approximates the output $f(x)$. $x_i$ achieves the top-most score ($a_i$) is regarded as the most important pixel for prediction, whereas those with the bottom-most score are considered least important.

With a given sorted $A^f(x) = \{a_i, i = 1...M | a_i < a_{i+1}\}$ generated by a typical feature attribution method, if the prediction f(x) can be approximated with the sum of $N$ most important features and $N$ is much less than $M$, we name such distribution of $A^f(x)$ is unequal. Namely, the prediction on $x$ made by model $f$ is dominated by a few pixels. Formally, we use Gini index (Dorfman, 1979) to measure the inequality of the distribution of a given feature attribution map. Given a population set indexed in non-decreasing order $\Phi = \{\phi_i, i = 1...n | \phi_i \leq \phi_{i+1}\}$, Gini coefficient can be calculated as:

$$Gini(\Phi) = \frac{1}{n} \left( n + 1 - 2\frac{\sum_{i=1}^n (n + 1 - i) * \phi_i}{\sum_{i=1}^n \phi_i} \right) \tag{4}$$

An advantage of the $Gini(\cdot)$ index is that inequality of the entire distribution can be summarized by using a single statistic that is relatively easy to interpret (see Appendix A.3 for a more detailed comparison between Gini and other sparsity measures). The Gini index ranges from 0, when the value of every $\phi_i$ is equal, to 1, when a single $\phi_i$ takes all the sum. This allows us to compare the inequality degree among feature attributions with different sizes. We define two types of inequality as follows:

- **Global inequality:** Given a feature attribution map $A^f(x) = \{a_i, i = 1...M | a_i < a_{i+1}\}$ on test data point $x$, we only consider the inequality degree of the global distribution of $A^f(x)$ and take no into account for other factors, the inequality degree is calculated with $Gini_g(A^f(x))$ directly. The higher of $Gini_g(A^f(x))$, the more unequal the distribution $A^f(x)$, the fewer pixels take the most prediction power. When $Gini_g(A^f(x))$ is equal to 1, it indicates one pixel dominates the prediction while all the other pixels have no contribution to the current prediction.

- **Regional inequality:** We also consider inequality degree together with spatial factor, whether important feature tends to cluster at specific regions. A region is defined as a block with size of $n * n$ of input space. We first divide pixels into different regions and calculate the sum of pixels' importance by regions, formally, $A_r^f(x) = \{a_{r_i}, i = 1...m | a_{r_i} < a_{r_{i+1}}\}$, where $a_r$ is the sum of $a_i$ in the region. Therefore, the Gini value on $A_r^f(x)$ reflects the inequality degree of different regions of input space. The higher the value of $Gini_r(A_r^f(x))$, the more important pixels tend to cluster in the specific regions. When $Gini_r(A_r^f(x))$ is equal to 1, it represents all pixels that contribute to the prediction cluster in one region (block).

In what follows, we propose potential threats caused by such inequality (global and regional inequality). We devise attacks utilizing common corruptions to reveal the unreliability of such decision pattern by $l_\infty$-robust model.

### 3.2 ATTACK ALGORITHMS

We propose two simple attacks to validate potential threats caused by such inequality phenomena: 1) Inductive noise attack. 2) Inductive occlusion attack.

#### 3.2.1 INDUCTIVE NOISE ATTACK

We evaluate the models' performance under attacks designed by two types of noise.

- **Noise (Type I):** Given an image $x$, we perturb the most influential pixels of images with Gaussian noise $\sigma \in \mathcal{N}(0, 1)$ via masking $M$. Formally:

$$x' = x + M * \sigma, \quad where \quad M_i = \begin{cases} 0, a_i < a_{tre} \\ 1, a_i \geq a_{tre} \end{cases} \tag{5}$$

  where $a_{tre}$ represents the threshold. $x_i$ with value that is below to the $a_{tre}$ will be kept, and $x_i$ whose $a_i \geq a_{tre}$ is perturbed by Gaussian noise.
- **Noise (Type II):** About the second type of noise attack, we directly replace important pixels with Gaussian noise, formally $x' = \overline{M} * x + M * \sigma$, where $\overline{M}$ represents reverse mask of $M$. Compared to Noise-I, Noise-II replaces important pixels totally and disturbs images more severely.

If the model's decision pattern is extremely unequal, the performance will be highly influenced when important features are corrupted by inductive noise attacks.

#### 3.2.2 INDUCTIVE OCCLUSION ATTACK

With respect to inductive occlusion attack, we obscure regions of important pixels with occlusions gradually. During the attack, the max count of occlusions is $N$ with a radius at max $R$. The order of regions to perturb is decided by the value of $A_r^f(x)$, that region of higher $a_{r_i}$ is perturbed in priority by occlusions with size $r \in \{1...R\}$. The number of occlusions is constrained by $n \in \{1...N\}$. We also consider occlusion with different colors to reflect potential real-world occlusion. The inductive occlusion attack algorithm is listed as follows:

---
**Algorithm 1** Inductive Occlusion Attack

---
**Require:** Test data point $(x, y)$, Model $f$, Regional Attribution map $A_r^f(x)$, Max count and radius $N, R$, Perturb color $c$.
**Ensure:** $f(x) = y$            ▷ Ensure the test data $x$ is correctly classified by model $f$.
     $n \leftarrow 1, r \leftarrow 1, x' = x$
     **for** $n = 1$ to $N$ **do**
         **for** $r = 1$ to $R$ **do**
             $M \leftarrow$ get_perturb_ mask($A_r^f(x)$, n, r)     ▷ A function to acquire the perturbation mask.
             $x' = \overline{M} * x + M * c$            ▷ Perturb $x$ by mask $M$ with color $c$.
             **If** $f(x') \neq y$ :**break**
         **end for**
     **end for**
     **return** $x'$

---

Note the intention of this work is not to propose strong adversarial attacks. Although either noise or occlusion is beyond the threat model considered in $l_\infty$ adversarial training, we intend to reveal the threats caused by such inequality phenomena that previous work ignored. In summary, the extreme inequality decision pattern of $l_\infty$-trained adversarial models to result in themselves being more fragile under some corruptions.

## 4 EXPERIMENTS

In this section, we first outline the experimental setup. We then evaluate the inequality degree (by Gini) of different models. Then, we evaluate the performance of the proposed attacks. Finally, we perform an ablation study about the selection of feature attribution methods.

## 4.1 EXPERIMENTAL SETTINGS

**Dataset and models.** We perform a series of experiments on ImageNet Deng et al. (2009) and CI-FAR10 Krizhevsky et al. (2009). With respect to experiments on ImageNet, we use ResNet18 (He et al., 2016), ResNet50, WideResNet50 (Zagoruyko & Komodakis, 2016) provided by Microsoft [1]. For CIFAR10, we use ResNet18, DenseNet (Huang et al., 2017) (see A.1 for detailed configurations). Regarding feature attribution methods (implementation by Captum[2]), we consider methods including Input X Gradients (Shrikumar et al., 2016), Integrated Gradients (Sundararajan et al., 2017), Shapley Value (Lundberg & Lee, 2017) and SmoothGrad (Smilkov et al., 2017). Considering space and time efficiency, we primarily present experimental results based on Integrated Gradients and perform an ablation study on the other feature attribution methods.

**Metrics.** For all the tests about the models' performance, we use error rate (%) as the metric to evaluate the model's performance under corruptions (e.g., noise and occlusions), which is the proportion of misclassified test images among the total number of test images defined as $\frac{1}{N}\sum_{n=1}^{N}[f(x) \neq f(x')]$, where $x$ represents clean test images, and $x'$ represents test images corrupted by noise and occlusions. For a fair comparison, we first select 1000 random images from ImageNet that are correctly classified by all the models before performing the attack.

## 4.2 INEQUALITY TEST

In this section, we first evaluate the inequality degree (both global and regional inequality) of $l_{\infty}$-robust models and standard trained models with different architectures (ResNet18, ResNet50, WideResNet, DenseNet) trained on ImageNet and CIFAR10. We also evaluate the inequality degree of different models adversarially trained with increasing adversarial strength ($\epsilon = 1, 2, 4, 8$). In the case of the evaluation on Gini, We applied the Gini index to the sorted absolute value of the flattened feature attribution maps. On evaluating regional inequality, we set the region's size as $16 * 16$ for experiments on ImageNet and $4 * 4$ for CIFAR10. The results are presented in Table 1. As shown

Table 1: **Gini index across different models.** We evaluate the Gini coefficient of different models trained with different $\epsilon$ on ImageNet and CIFAR10.

| Dataset | Model | Std. trained | $\epsilon = 1.0$ | $\epsilon = 2.0$ | $\epsilon = 4.0$ | $\epsilon = 8.0$ |
|---------|-------|--------------|------------------|------------------|------------------|------------------|
| | | **Global Inequality** | | | | |
| **CIFAR10** | **ResNet18** | $0.58 \pm 0.05$ | $0.65 \pm 0.05$ | $0.67 \pm 0.06$ | $0.69 \pm 0.06$ | $0.73 \pm 0.06$ |
| | **DenseNet** | $0.57 \pm 0.04$ | $0.66 \pm 0.06$ | $0.67 \pm 0.06$ | $0.69 \pm 0.06$ | $0.72 \pm 0.07$ |
| | | **Regional Inequality** | | | | |
| | **ResNet18** | $0.79 \pm 0.02$ | $0.87 \pm 0.04$ | $0.87 \pm 0.04$ | $0.88 \pm 0.04$ | $0.88 \pm 0.04$ |
| | **DenseNet** | $0.79 \pm 0.02$ | $0.85 \pm 0.04$ | $0.86 \pm 0.04$ | $0.87 \pm 0.04$ | $0.88 \pm 0.03$ |
| | | **Global Inequality** | | | | |
| **ImageNet** | **ResNet18** | $0.60 \pm 0.04$ | $0.69 \pm 0.06$ | $0.79 \pm 0.04$ | $0.92 \pm 0.01$ | $0.95 \pm 0.01$ |
| | **ResNet50** | $0.62 \pm 0.04$ | $0.75 \pm 0.05$ | $0.86 \pm 0.03$ | $0.92 \pm 0.02$ | $0.94 \pm 0.01$ |
| | **WideResNet** | $0.62 \pm 0.05$ | $0.74 \pm 0.05$ | $0.79 \pm 0.04$ | $0.88 \pm 0.03$ | $0.94 \pm 0.01$ |
| | | **Regional Inequality** | | | | |
| | **ResNet18** | $0.80 \pm 0.02$ | $0.83 \pm 0.04$ | $0.88 \pm 0.03$ | $0.95 \pm 0.01$ | $0.97 \pm 0.01$ |
| | **ResNet50** | $0.84 \pm 0.02$ | $0.91 \pm 0.05$ | $0.95 \pm 0.02$ | $0.96 \pm 0.01$ | $0.97 \pm 0.01$ |
| | **WideResNet** | $0.81 \pm 0.03$ | $0.86 \pm 0.03$ | $0.88 \pm 0.03$ | $0.93 \pm 0.03$ | $0.97 \pm 0.02$ |

in Table 1, on CIFAR10, the global inequality degree of the standard trained model with different architectures is around 0.58. The Gini (global inequality) for $l_{\infty}$-robust model is around 0.73 when $\epsilon = 8$. Notably, the inequality phenomena is much more obvious on ImageNet. Especially for an adversarially trained Resnet50 ( $\epsilon = 8$), the Gini achieves 0.94, which indicates that only a handful of pixels dominate the prediction. Experiments on CIFAR10 and ImageNet show that $l_{\infty}$-robust models rely on fewer pixels to support the prediction with the increasing of the adversarial strength

---

[1]https://github.com/microsoft/robust-models-transfer

[2]https://github.com/pytorch/captum

($\epsilon$). We also test the inequality degree on different classes belonging to ImageNet; classes related to animal tends to have a higher value on Gini index. For example, class 'Bustard' has the highest value on Gini of 0.950. Classes related to scenes or stuff tend to have a lower Gini. For example, class 'Web site' has the lowest inequality of 0.890 (See Appendix A.7).

We visualize the features' attribution of given images for the standard and $l_\infty$-adversarially trained ResNet50 respectively in Figure 2. When the model is adversarially trained with weak adversarial

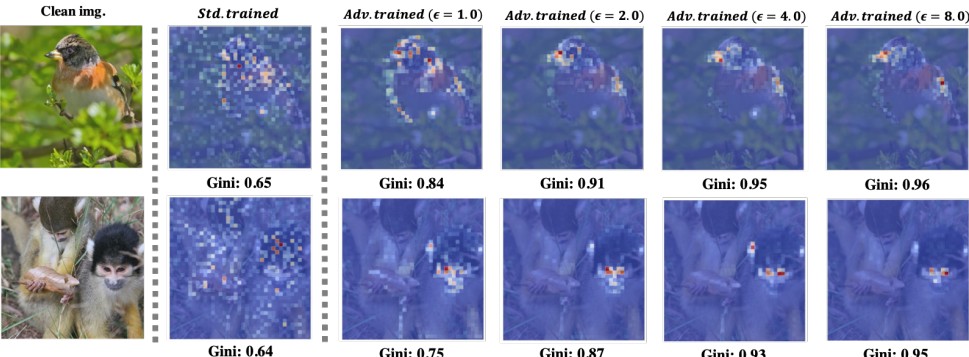

Figure 2: **Feature attributions of different models.** We visualize feature attributions generated by $l_\infty$-robust models (adversarially trained by adversaries of different $\epsilon$), the larger of $\epsilon$, the fewer features that model relies on for prediction.

strength ($\epsilon = 1$), the model has better feature attribution aligned to human observers. However, when the adversarial strength increases, the model gradually assigns higher importance to fewer pixels and resulting in extreme inequality regarding feature attribution. Moreover, these most important pixels tend to gather in a few specific regions ( Additional visualizations for ImageNet and CIFAR10 are in Appendix A.11 and A.10 respectively).

### 4.3 EVALUATION UNDER INDUCTIVE NOISE ATTACK

In this part, we compare the performance of standard- and adversarially- trained ResNet50 under random and inductive noise. We set noise with different scales, including subpixels of 500, 1000, 5000, 10000, and 20000. We present the results in Figure 3.

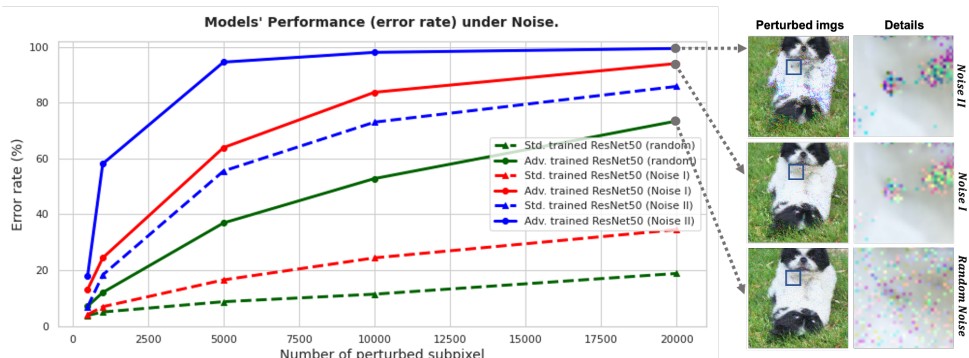

Figure 3: **Evaluation under noise.** We plot the error rate of standard- and adversarially-trained models on images perturbed by the increasing number of noise.

Under random noise, the success rate of attack on the robust model achieves 73.4%, but only 18.8% for standard-trained model. Under Noise of Type I, the robust model is fooled by 94.0%, while the standard trained model is only fooled by 34.5%. Under Noise of Type II, even when we control the amount of noise with a small threshold (e.g., 1000 pixels), more than 50% of predictions made by the robust model is affected. When we enlarge the threshold to 20000, the robust model ($\epsilon=8$) is almost fooled with a 100% success rate. In summary, compared to the standard trained model,

$l_\infty$-robust model relies on much fewer pixels to make a decision; such a decision pattern results in unstable prediction under noise.

### 4.4 EVALUATION UNDER INDUCTIVE OCCLUSION ATTACK

In this part, we perform an inductive occlusion attack and evaluate the standard- and $l_\infty$-robust ResNet50s' performance. We set two group experiments with different thresholds.

In the first group of experiments, we generate occlusions with a max count of 5 and a max radius of 10. In this case, the adversarially-trained model is fooled at a 71.7% error rate, but the standard trained model' predictions are only affected by 31.6%. When we enlarge the threshold and set max count as 10 and radius as 20 for occlusions, both $\epsilon = 4$ and $\epsilon = 8$ adversarially trained model can be fooled with 100% success rate while only 41.2% attack success rate for the standard-trained model. We visualize the results in Figure 4. As the figure shows, $l_\infty$-adversarially trained model with larger $\epsilon$ could be easily attacked by smaller occlusions even under the same threshold. For example, in Figure 4, the standard trained model can recognize 'Bulbul' well with the head part occluded, but the adversarially trained model fails to recognize the 'Bulbul' if only the beak of the bulbul is occluded. Moreover, compared with adversarial perturbation, occlusion is more practi-

Table 2: **Models' performance (Error rate %) under occlusions.** We evaluate the models' performance by gradually occluding important areas with patches of different sizes and colors.

| Model | Std. | $\epsilon = 1.0$ | $\epsilon = 2.0$ | $\epsilon = 4.0$ | $\epsilon = 8.0$ |
|---|---|---|---|---|---|
| **Max cnt N = 5, R = 10** | | | | | |
| Occlusion-G | 23.5% | 31.6% | 38.4% | 32.4% | 54.0% |
| Occlusion-W | 28.3% | 48.4% | 57.5% | 61.3% | 71.7% |
| Occlusion-B | 31.6% | 51.5% | 53.3% | 48.9% | 64.6% |
| **Max cnt N = 10, R = 20** | | | | | |
| Occlusion-G | 30.1% | 48.2% | 56.3% | 100.0% | 100.0% |
| Occlusion-W | 40.1% | 59.1% | 73.3% | 100.0% | 100.0% |
| Occlusion-B | 41.2% | 70.2% | 72.2% | 100.0% | 100.0% |

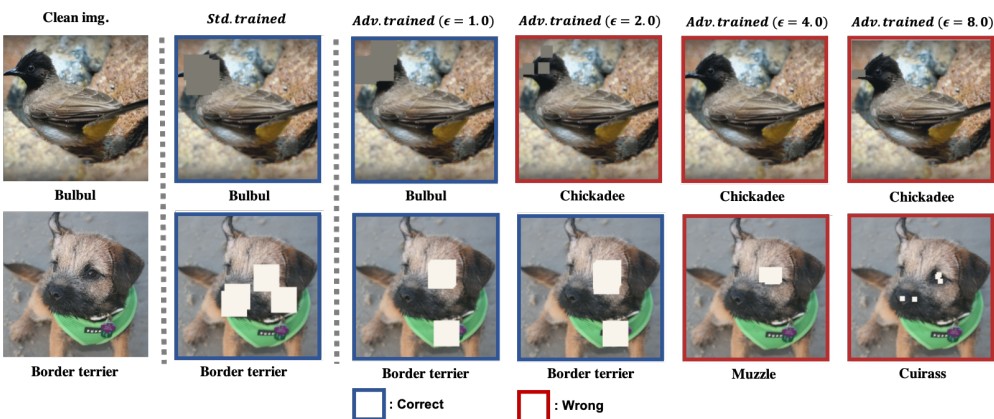

Figure 4: **Visualization of occluded images.** We visualize images occluded with different patches of different sizes and the corresponding predictions made by standard and $l_\infty$-adversarially trained models. Compared to a standard-trained model, the adversarially trained model is fragile when occlusion covers the area of important features.

cal as occlusion frequently appears in the real world. We also evaluate the transferability of attacked results between the robust model and the standard trained model, the results are consistent with our observation (see Appendix A.4).

## 4.5 Ablation Study

We consider four attribution methods for the ablation study: Input X Gradient (Shrikumar et al., 2016), SmoothGrad (Smilkov et al., 2017), Gradient Shapley Value (GradShap for short) (Lundberg & Lee, 2017) and Integrated Gradients (Sundararajan et al., 2017) (see Appendix A.8 for detailed configuration). We perform an ablation study to evaluate the effect of selection on the feature attribution methods (see Table 3). Among various attribution methods, SmoothGrad produces more

Table 3: **Ablation study on selection.** We evaluate our hypothesis with different feature attribution methods.

| Attribution Method | Model | Gini | Noise I | Noise II | Occlusion-B | Occlusion-G | Occlusion-W |
|---|---|---|---|---|---|---|---|
| **Input X Gradient** | Std. trained | 0.63 | 16.1% | 45.0% | 24.7% | 16.8% | 23.5% |
| | Adv. trained | 0.93 | 60.9% | 90.4% | 63.3% | 51.2% | 63.2% |
| **GradShap** | Std. trained | 0.62 | 19.8% | 54.7% | 31.5% | 24.1% | 29.9% |
| | Adv. trained | 0.93 | 62.3% | 93.7 % | 64.8% | 53.3% | 71.6% |
| **SmoothGrad** | Std. trained | 0.75 | 63.5% | 45.0% | 32.3% | 25.6% | 30.3% |
| | Adv. trained | 0.98 | 82.5% | 98.3% | 61.6% | 49.7% | 60.8% |
| **Integrated Gradients** | Std. trained | 0.62 | 16.5% | 55.5% | 31.6% | 23.5% | 28.3% |
| | Adv. trained | 0.94 | 63.9 % | 95.5% | 64.6% | 54.0% | 71.7% |

spare feature attribution maps and thus results in higher values on Gini. Regarding evaluation under noise, SmoothGrad increases the inductive noise attack's success rate. Regarding evaluation under occlusion, the selection of Integrated Gradients improve the attack's success rate on models.

In conclusion, the selection of attribution methods slightly affects attacks' success rates but does not change our conclusion: the distribution of features' attribution by $l_\infty$-robust model is much more unequal; such inequality makes the robust model more susceptible to inductive noise and occlusions.

## 5 Discussion and Conclusion

In this work, we study the inequality phenomena that occur during $l_\infty$-adversarial training. Specifically, we find $l_\infty$-robust models' feature attribution is not as aligned with human perception as we expect. An ideal human-perceptual aligned model is expected to make decisions based on a series of core feature attributions. For example, if the model classifies an input image as a bird, it should take attributions, including the eye, the beak of the bird, and the shape of the bird, all of these attributions into account. However, we find $l_\infty$-robust model only relies on individual attribution (only the bird's beak) for recognition. We name such phenomena as inequality phenomenon. We perform extensive experiments to evaluate such inequality phenomena and find that $l_\infty$ robust model assigns a few features with extremely high importance. Thus, a few features dominate the prediction. Such extreme inequality of $l_\infty$-robust model results in unreliability. We also design attacks (by utilizing noise and occlusion) to validate our hypothesis that robust models could be more susceptible under some scenarios. We find an attacker can easily fool the $l_\infty$-trained model by modifying important features with either noise or occlusion easily. We suggest that both noise and occlusion are common in a real-world scenario. Therefore, robustness against either noise or occlusion is more essential and crucial than robustness against adversarial examples. Our work reveals the limitation and vulnerability of the current $l_\infty$-robust model. We also evaluate if such inequality phenomenon exists in $l_2$-robust model and models trained with sparsity regularization. The evaluation results show that such a phenomenon is a unique property of $l_\infty$-robust model (see Appendix A.5 and A.6).

We also propose a strategy to release such inequality phenomena during $l_\infty$-adversarial training. We combine Cutout (DeVries & Taylor, 2017) strategy with adversarial training and force the model learning features from different regions by cutting out part of training images at each iteration during the training (see the result in Appendix A.9). The strategy slightly releases the inequality degree of the robust model. More effective strategies releasing such extreme inequality could be a crucial and promising direction for future work. We hope our work can motivate new research into the characteristics of adversarial training and open up further challenges for reliable and practical adversarial training.

ETHICS STATEMENT

In this paper, we identify inequality phenomena that occur during $l_\infty$-adversarial training, that $l_\infty$-robust model tends to use few features to make the decision. We give a systematical evaluation of such inequality phenomena across different datasets and models with different architectures. We further identified unrealized threats caused by such decision patterns and validated our hypothesis by designing corresponding attacks. Our findings provide a new perspective on inspecting adversarial training. Our goal is to understand current adversarial training's weaknesses and make DNNs truly robust and reliable. We did not use crowdsourcing and did not conduct research with human subjects in our experiments. We cited the creators when using existing assets (e.g., code, data, models).

REPRODUCIBILITY STATEMENT

We present the settings of hyper-parameters and how they were chosen in the experiment section. We repeat experiments multiple times with different random seeds and show the corresponding standard deviation in the tables. We plan to open the source code to reproduce the main experimental results later.

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

# A APPENDIX

## A.1 DETAILS OF MODELS

### A.1.1 DETAILS OF MODELS TRAINED ON IMAGENET

We summarize the clean and robust accuracy (%) of models trained on ImageNet in Table **??**. Regarding robust accuracy, we use PGD for evaluation. During the training on ImageNet, the images

Table 4: $l_\infty$-robust models' clean and robust accuracy (ImageNet)

| Model | Std. trained | $\epsilon$=1/255 | $\epsilon$=2/255 | $\epsilon$=4/255 | $\epsilon$=8/255 |
|---|---|---|---|---|---|
| ResNet-18 | 69.76 | 63.46 /33.38 | 59.63 /29.80 | 52.49 /26.52 | 42.11 /23.75 |
| ResNet-50 | 76.13 | 72.05 /45.02 | 69.10 / 42.75 | 63.86 / 38.85 | 54.53 / 33.05 |
| Wide-ResNet-50 | 81.60 | 74.65 /47.30 | 72.35 /45.95 | 68.41 /43.10 | 60.82 /38.35 |

are resized to 256 using interpolation=InterpolationMode.BILINEAR, followed by a central crop of size=224. Finally, the values are first rescaled to [0.0,1.0] and then normalized using mean=[0.485, 0.456, 0.406] and std=[0.229, 0.224, 0.225].

```
TRAIN_TRANSFORM = transforms.Compose([
    transforms.Resize(256),
    transforms.CenterCrop(224),
    transforms.RandomHorizontalFlip(),
    transforms.ToTensor(),
])
TEST_TRANSFORM = transforms.Compose([
    transforms.Resize(256),
    transforms.CenterCrop(224),
    transforms.ToTensor(),
])
```

### A.1.2 DETAILS OF MODELS TRAINED ON CIFAR10

We summarize the clean and robust accuracy of models trained on CIFAR10 in Table 5. Regarding robust accuracy, we use AutoAttack Croce & Hein (2020) for evaluation.

Table 5: $l_\infty$-robust models' clean and robust accuracy (CIFAR10)

| Model | Std. trained | $\epsilon$=1/255 | $\epsilon$=2/255 | $\epsilon$=4/255 | $\epsilon$=8/255 |
|---|---|---|---|---|---|
| **ResNet-18** | 93.90 | 92.10 /86.60 | 90.40 /79.90 | 88.30 /68.80 | 81.20 /48.60 |
| **DenseNet** | 92.80 | 91.30 /86.60 | 89.90 /79.50 | 85.80 /65.90 | 79.60 /44.40 |

```
TRAIN_TRANSFORM = transforms.Compose([
    transforms.RandomCrop(32, padding=4),
    transforms.RandomHorizontalFlip(),
    transforms.ToTensor(),
])

TEST_TRANSFORM = transforms.Compose([
    transforms.ToTensor(),
])
```

A.2    FURTHER DISCUSSION ABOUT VISUALIZATION OF FEATURE ATTRIBUTION

Most visualization methods apply post-processing techniques during generating feature attribution maps. The post-processing technique is also clarified in (Tsipras et al.): "For CIFAR-10 and ImageNet, we clip gradients to within $\pm 3\sigma$ and rescale them to lie in the [0, 1] range." Thus, the most influential pixels with extremely high values are clipped to a relatively lower value but they actually dominate the prediction (see Figure 5).

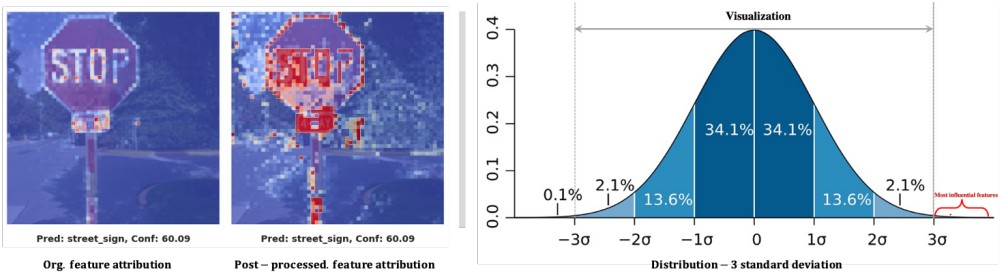

Figure 5: **Visualizing feature attribution with and without post-processing (by 3 deviations).**

We also provide more visualization of feature attribution maps with and without post-processing. As Figure 6 shows, the post-processed feature attribution maps are more perceptually-aligned with human observers. However, such visualization are not subjective.

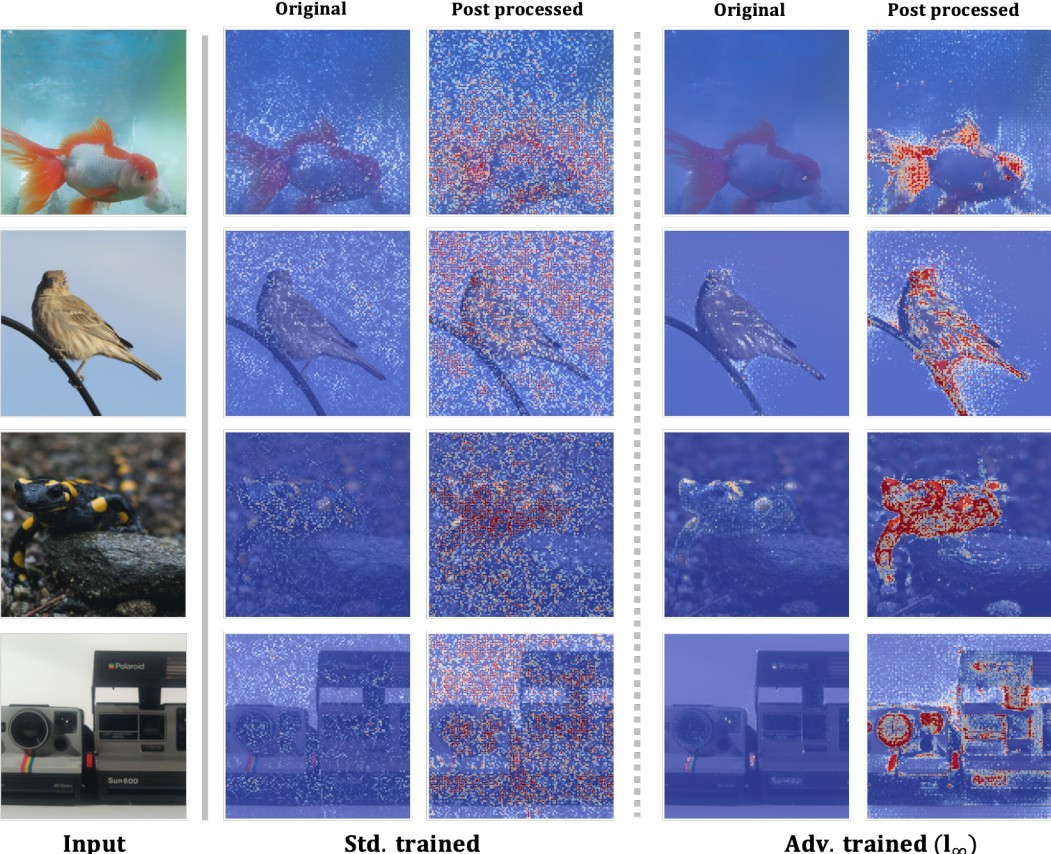

Figure 6: **More visualization of feature attribution maps with and without post-processing** .

## A.3 Sparsity measure vs. Gini index & Motivation behind using Gini index

The sparsity of a vector can be quantified by $\frac{||x||_0}{|x|}$ (duf, 2017), which simply calculates the ratio of non-zero elements. However, the sparsity measure treats an infinitesimally small value the same as a significant value. Even if some of the small coefficients increase by significant values, that change will not be reflected by a change in the value of the sparsity measure. For example, given a vector $x_1 = [0, 0, 0, 1, 1]$ and $x_2 = [0, 0, 0, 1, 1000]$, the sparsity degree of $x_1$ and $x_2$ is equal to 0.4. However, their distributions are totally different. The distribution of $x_2$ is much more unequal compared with $x_1$. Also, $\frac{||x||_0}{|x|}$ is sensitive to noise, especially in settings where most values of elements are around 0 (e.g., feature attribution map).

In our case, Gini is able to reflect the change when a small coefficient increases. A Gini coefficient of 0 expresses perfect equality, where all values are the same, while a Gini coefficient of 1 (or 100%) expresses maximal inequality among values. For example, the Gini of $x_1 = [0, 0, 0, 1, 1]$ equals to 0.6, and 0.799 for $x_2 = [0, 0, 0, 1, 1000]$. The value of Gini also provides an intuitive understanding of the distribution. When Gini = 0.6, approximately 40% in the population (1-0.6 = 0.4) occupies the total worth. When Gini = 0.799, approximately 21.1% of the population dominates the worth. As for our experiment, Gini of feature attributions by $l_\infty$-robust model ($\epsilon = 8.0$) is about 0.95, representing less than 5% of pixels that dominate the prediction.

## A.4 Transferability test

we perform occlusion attacks with two groups of attack budgets:

- **Group 1**: max count = 5, max radius = 10.
- **Group 2**: max count = 10, max radius = 20.

We perform noise attacks with threshold = 5000.

| Attack | Occ -B (cnt=5, r=10) | | Occ-B (cnt=10, r=20) | | Noise-I | | Noise-II | |
|---|---|---|---|---|---|---|---|---|
| **Model** | **Adv.** | **Std.** | **Adv.** | **Std.** | **Adv.** | **Std.** | **Adv.** | **Std.** |
| **Adv.** | 100.0 | 11.4 | 100.0 | 17.4 | 100.0 | 10.6 | 100.0 | 18.6 |
| **Std.** | 22.8 | 100.0 | 43.6 | 100.0 | 26.8 | 100.0 | 58.2 | 100.0 |

As the table shows, the transferability between the $l_\infty$-robust model and the standard-trained model is low. Transferring attack results from the standard-trained model to the $l_\infty$-robust model is easier. If the region of the most important pixels is occluded, the $l_\infty$-robust model fails to recognize the images correctly. The experiments are consistent with our observations.

## A.5 Compared with $l_2$-robust model

Due to different properties of $l_p$ norm vector space, the case of $l_2$ adversarial training is not the same as $l_\infty$ adversarial training. To be specific, $l_\infty$ constrains the maximum magnitude of perturbation for each pixel. The adversarial noise is added on each pixel independently during the $l_\infty$ adversarial training. Therefore, the model attempts to find the most robust feature against noise and drops the features which could be affected by adversarial noise. Therefore, with increasing the magnitude ($\epsilon$) of adversarial noise, fewer but more robust features $l_\infty$-robust model can rely on for recognition.

Different from $l_\infty$ norm measures each pixel independently, $l_2$ norm calculates the square root of the inner product of all elements in a vector. Thus, during $l_2$ adversarial training, if a large budget of perturbation perturbs some pixels, the other pixels share the left budget on perturbation. Moayeri et al. provides a game-theoretic understanding of $l_2$-adversarial training: during each loop of $l_2$-adversarial training, the attacker perturbs the features which are predictive for the model. When some predictive features are perturbed with the most budget of perturbation, the features perturbed with small or without perturbation are easier for the model to learn. Furthermore, these less perturbed features then become more predictive at the next training iteration. Thus, the inequality phenomenon does not occur during $l_2$-adversarial training. However, the $l_2$-robust model would use both the object-relevant and object-irrelevant features (e.g., background) for prediction at the

game's equilibrium. In (Moayeri et al.; 2022), they show that $l_2$-robust model is more sensitive to the background and other spurious features.

| Model | Clean acc | Gini-g | Gini-r | Occ-B | Occ-G | Occ-W | Noise-I | Noise-II |
|---|---|---|---|---|---|---|---|---|
| **Std. trained** | 76.13% | 0.62 | 0.84 | 33.0% | 28.1% | 36.5% | 15.8% | 49.8% |
| **l2-AT model** | 56.13% | 0.60 | 0.76 | 51.4% | 32.5% | 48.3% | 2.9% | 36.3% |
| **linf-AT model** | 54.53% | 0.95 | 0.97 | 81.5% | 61.0% | 77.2% | 75.6% | 96.1% |

As explained above, the inequality degree of $l_2$-robust model is similar to the standard trained model (or $l_2$ is even more equal). However, $l_2$-robust model is still more vulnerable to occlusion. We guess it is because the most influential features tend to cluster together for both $l_\infty$ at and $l_2$-robust model.

## A.6 COMPARED WITH MODELS TRAINED WITH SPARSITY REGULARIZATION

we perform experiments on $l_\infty$-robust model and model trained with regularization for sparsity. We consider two types of sparse models: the model with sparse architecture and models with sparse weights. We compared $l_\infty$-robust model with models regularized by the following techniques:

- $l_1$ norm pruning (Li et al.): It prunes filters by removing whole filters in the network together with their connecting feature maps.
- Weight pruning (Han et al., 2015): It applies mask as regularization and sets the weights to be 0. It results in sparser weights and connectivity patterns.

| Model | Clean Acc | Gini g | Gini r | Occ-B | Occ-G | Occ-W | Noise-I | Noise_II |
|---|---|---|---|---|---|---|---|---|
| **Adv. trained** | 54.53% | 0.94 | 0.97 | 78.2% | 80.2% | 60.2% | 60.9% | 95.1% |
| **L1-norm sparsity** | 73.08% | 0.60 | 0.80 | 56.8% | 57.8% | 52.5% | 40.7% | 89.2% |
| **Weight-level Pruning** | 75.60% | 0.62 | 0.81 | 33.1% | 29.2% | 25.8% | 15.7% | 49.9% |

The sparsity of either model architecture or weights will not result in inequality on feature attribution. The $l_\infty$-AT model is much more easily affected by occlusion and noise attack than the two sparse models. We think $l_\infty$ can be regarded as a strong regularization: during $l_\infty$-adversarial training, the model attempts to find the most robust feature against adversarial noise and discards the features which could be affected by added adversarial noise. With increasing the magnitude ($\epsilon$) of adversarial noise, only a handful of features $l_\infty$-adversarially trained model can rely on for recognition.

## A.7 EQUALITY DEGREE OF DIFFERENT CLASSES

We test the inequality degree of feature attributions' distribution of 50000 samples from 1000 classes in ImageNet. We present results in Table 6.

Table 6: **Global inequality degree of different classes** ($l_\in$-**Adv. traind**, $\epsilon = 8$)

| Top-5 | Class | Bustard | Manhole cover | Oystercatcher | Redshank | Pomeranian |
|---|---|---|---|---|---|---|
| | Gini | 0.950 | 0.949 | 0.949 | 0.949 | 0.949 |
| Bottom-5 | Class | Web site | Slot | Grocery store | Grille | Comic book |
| | Gini | 0.890 | 0.900 | 0.901 | 0.901 | 0.905 |

Classes with top-5 inequality (Gini value) are: Bustard, Manhole cover, Oystercatcher, Redshank and Pomeranian. And their Gini values are 0.950, 0.949, 0.949, 0.949 and 0.949. Classes with bottom-5 inequality (Gini value) are: Web site, Slot, Grocery store, Grille and Comic book. Their corresponding Gini values are: 0.890, 0.900, 0.901, 0.901 and 0.905.

Regarding regional inequality, we present results in Table 7. Classes with high regional inequality are similar to classes with high global inequality. Specifically, classes of the top 5 (regional inequality) are Redshank, American coot, Oystercatcher, Bustard and Gazelle. And the $Gini_r(.)$ of these

Table 7: **Regional inequality degree of different classes ($l_\in$-Adv. traind, $\epsilon = 8$)**

| Top-5 | Class | Redshank | American coot | Oystercatcher | Bustard | Gazelle |
|---|---|---|---|---|---|---|
| | Gini | 0.976 | 0.975 | 0.975 | 0.974 | 0.974 |
| Bottom-5 | Class | Web site | Grocery store | Confectionery | Slot | Grille |
| | Gini | 0.929 | 0.932 | 0.934 | 0.935 | 0.936 |

classes are: 0.976, 0.975, 0.975, 0.974 and 0.974. Classes in the bottom 5 (regional inequality) are Web site, Grocery store, Confectionery, Slot, and Grille. And the $Gini_r(.)$ of these classes are: 0.929, 0.932, 0.934, 0.935 and 0.936.

## A.8 DETAILS ABOUT FEATURE ATTRIBUTION METHOD

We use attribution methods including Input X Gradient, Smooth Gradient (short for SmoothGrad), Gradient Shapley Value (short for GradShap) and Integrated Gradients.

**Input X Gradient:** The Input X Gradient multiplies input with the gradient with respect to each input feature. It is a baseline approach for computing the attribution.

**GradShap:** Shapley Values aims compute each feature's attribution based on cooperative game theory. GradShap approximates Shapley Values by computing the expectations of gradients by randomly sampling from the distribution of $baselines$. It adds noise to each input sample $n_{samples}$ times, selects a random baseline from the baselines' distribution, and a random point along the path between the baseline and the input. Then it computes the gradient of outputs with respect to those selected random points. In our evaluation, we set $n_{samples} = 20$ for experiments with ImageNet and $n_{samples} = 10$ for experiments with CIFAR10. We set $baseline = 0$ for all the experiments.

**Integrated Gradients:** Integrated Gradients is an axiomatic model which assigns an importance score to each input feature by approximating the integral of gradients of the model's output with respect to the inputs along the path (straight line) from given $baseline$ to inputs. Previous work points out Integrated Gradients method is sensitive to the choice of path. To reduce such sensitivity, Integrated Gradients are usually repeated for $n_{step}$ steps. For all the experiments, we set $n_{step} = 20$, and $baseline \in \mathcal{N}(0, 1)$.

**SmoothGrad:** SmoothGrad adds gaussian noise to each input in the batch $n_{samples}$ times, then applies the given attribution algorithm to each of the samples. It returns the mean of the sampled attributions. SmoothGrad returns a sparser feature attribution map than other methods. In our experiment, we set $n_{samples} = 20$

## A.9 HOW TO RELEASE INEQUALITY PHENOMENON

We also try to propose a strategy to release the inequality phenomenon in adversarial training. Intuitively, we hope adversarially trained models learn to find robust features from the whole image rather than focus on a specific robust feature. Towards this end, we incorporate Cutout with adversarial training that Cutout enables the model to learn features from multiple spatial spaces. We evaluate our strategy on CIFAR10. The results are presented in Table 8.

| Model | Gini | Gini-R | Clean Acc. | Adv. Acc. | Noise I | Noise II | Occlusion-B | Occlusion-G | Occlusion-W |
|---|---|---|---|---|---|---|---|---|---|
| **Adv. trained** | 0.73 | 0.88 | 82.10 % | 48.50 % | 40.31 % | 1.58% | 100.00% | 100.00% | 100.00% |
| **Adv. trained+Cutout** | 0.70 | 0.88 | 81.40 % | 47.20 % | 40.78% | 1.72% | 93.39% | 29.98% | 55.06 % |

Table 8: $l_\infty$-**adversarial training with and without Cutout.**

As the table indicates, the Cutout strategy can slightly release the inequality of $l_\infty$-adversarial training. But at a price, both clean accuracy and adversarial accuracy slightly decrease. Regarding performance under noise and occlusion, the strategy does not improve the adversarial trained model's performance under noise but improves its performance under occlusions. A more effective strategy to release inequality phenomena is highly required.

## A.10 VISUALIZATION FOR CIFAR10

We visualize feature attribution maps and attack results for CIFAR10. About the setting of attack, we set max count $N = 10$, with max radius $R = 4$. With occlusion with different colors (black, white and grey), success rates on 1000 correct classified images of $l_\infty$-adversarially trained model are 60.4%, 60.5%, and 38.1% respectively. And success rates for the standard trained models are 34.6 %, 36.7% and 24.1% respectively. We visualize corresponding results in Figure 8.

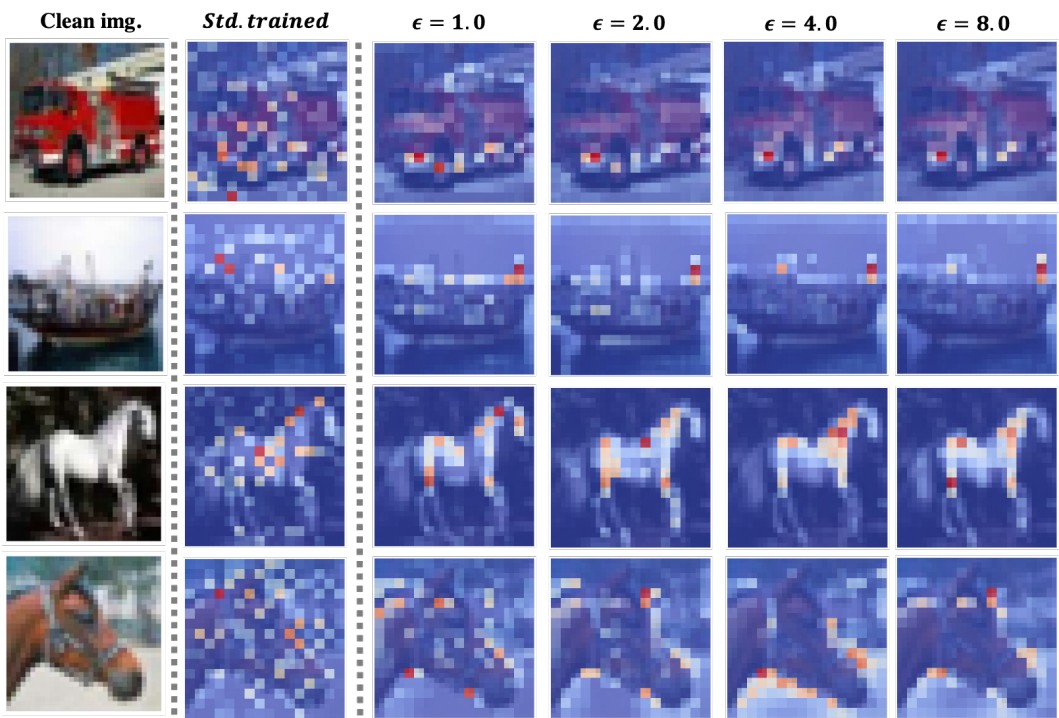

Figure 7: **Visualization of feature attribution on CIFAR10.**

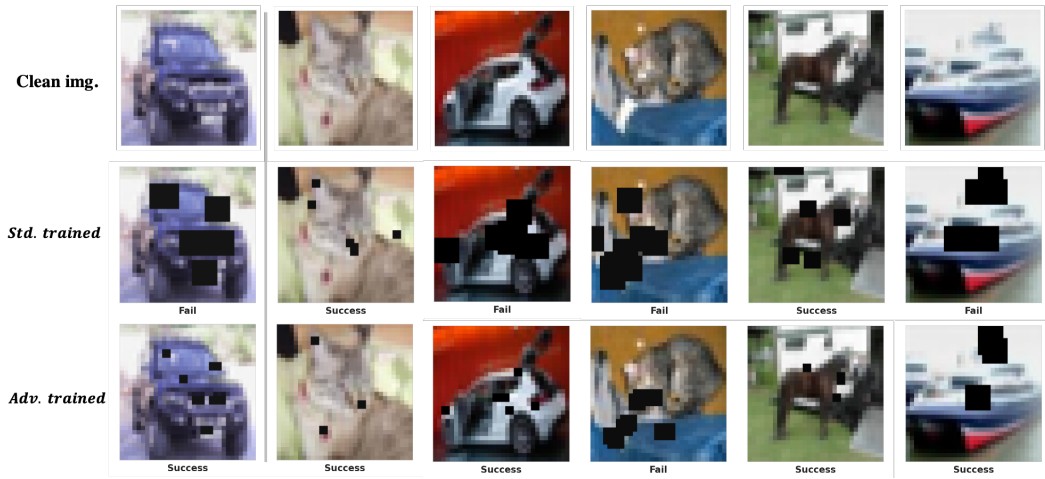

Figure 8: **Visualization of occluded images for CIFAR10.**

## A.11 MORE VISUALIZATION RESULTS FOR IMAGENET

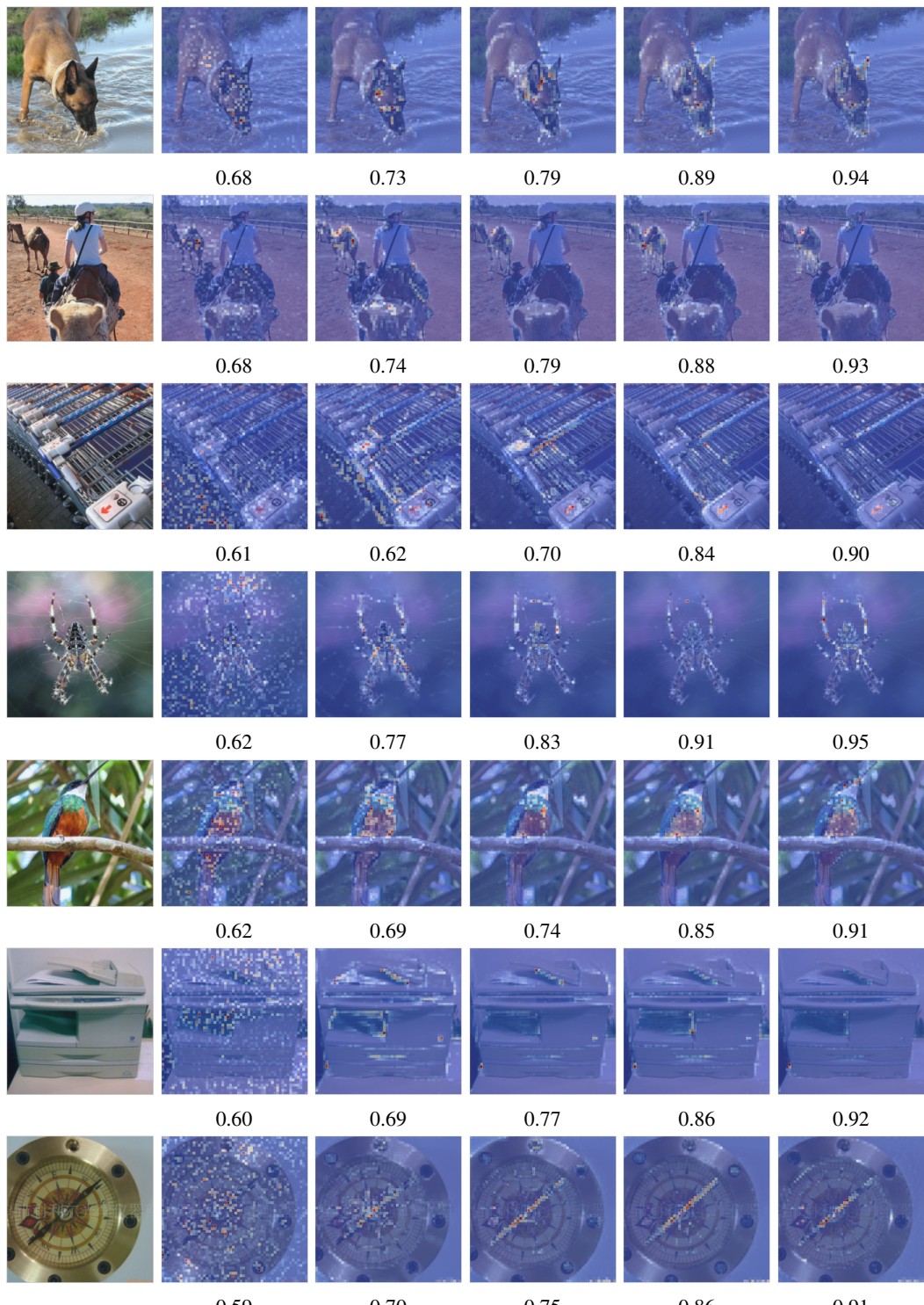

Figure 9: Visualization of feature attributions generated by standard- and adversarially trained model with different $\epsilon$.

