# OpenReview forum: "Inequality phenomenon in $l_{\infty}$-adversarial training, and its unrealized threats"
_ICLR.cc/2023/Conference — ICLR 2023 notable top 25%_

### Official Review · Reviewer_1FTk · 2022-10-18

**Confidence:** 4
**Correctness:** 4
**Technical Novelty And Significance:** 3
**Empirical Novelty And Significance:** 4
**Recommendation:** 8

**Clarity, Quality, Novelty And Reproducibility:**

Overall, I think that the paper's clarity can be greatly improved (see Weaknesses).  I think that overall the experiments are well-designed and novel and results are significant, but I think that the use of Gini index can be motivated a little better (as opposed to just using sparsity as in previous works).  I also think that additional baselines outside of just standard trained models should be used: specifically models trained with sparsity regularization should be considered.  Additionally, comparisons to L2 adversarially trained models can be interesting as well.

**Strength And Weaknesses:**

Strengths:
I think the experiments are interesting and highlight a weakness of Linf adversarial training well.

Weaknesses:
Paper clarity can be greatly improved, for instance the authors mention "inequality phenomenon" early in the paper, but it isn't clear what this means until after the authors describe using the Gini index. The authors should provide some short explanation of what they mean by "inequality phenomenon" early on in the text to improve clarity.  Some other parts I was confused about are
- at the beginning of Section 2.3 what is meant by "weak signals"
- in Section 3.1 "In detail, we treat the input image..." seems to be 2 separate sentences? or the second half of the sentence "feature attribution methods aim to assign..." should be deleted?
- In section 3.1, in the definition of regional inequality, what exactly does a region mean?  Is it a set of pixels in the input space?  How are these regions chosen?
- In 3.2.1, the text for the description of noise type 1 is confusing.

What is the motivation behind using Gini index?  How much more informative is Gini index compared to just measuring sparsity?  Does sparsity not imply that Gini index will increase?

Does this inequality phenomenon arise due to more sparse features or is a result of something more specific to Linf adversarial training.  How successful are these occlusion and noise attacks on models trained with regularization for sparsity?

All visualizations are on ImageNet.  It would be nice to see some visualizations for CIFAR-10 as well.

Could the authors provide clean and robust accuracies of models evaluated?

To confirm, the "error rate" is more of a measure of stability rather than error right?  If the model misclassifies the clean image and also misclassifies the perturbed image as the same class, then there is no error?  In that case, I think that the metric should be renamed to something involving "stability" rather than "error" to avoid confusion.  Another possible metric is to filter out the images that both the clean model and adversarially trained model are incorrect on and then measure error.  This would allow us to understand the frequency that a correctly classified image changes class which I think is a little more informative.

**Summary Of The Paper:**

This paper studies the negative impacts of sparse features as a result of Linf adversarial training.  The authors find that this sparsity causes the Linf robust models to be more vulnerable to small occlusions and smaller magnitudes of random Gaussian noise in comparison to standard trained models.

**Summary Of The Review:**

Overall, I think this is a very interesting work that investigates vulnerabilities that arise from Linf adversarial training.  The authors demonstrate these vulnerabilities by designing attacks against Linf adversarially trained models.  However, I think that there can be some improvements especially with regards to the clarity of the writing.  I also think that the authors should consider other baselines (specifically models trained with sparsity regularization) instead of just comparing to standard training.  This would allow us to better understand whether sparse features are the cause of these vulnerabilities or whether the observed vulnerabilities are unique to Linf adversarially trained models.

---

> ### Author Response · Authors · 2022-11-18
> **Response to Reviewer 1FTk (Part I)**
>
> **Q1. Paper clarity can be greatly improved.**
>
> > Thank you for your patience and careful attention to pointing out these issues typos.  We have fixed all of these typos in our revision.
>
> Q1.1. For instance the authors mention "inequality phenomenon" early in the paper, but it isn't clear what this means until after the authors describe using the Gini index. The authors should provide some short explanation of what they mean by "inequality phenomenon" early on in the text to improve clarity.
>
> > Thanks for your suggestion. We have added a short explanation of the "inequality phenomenon" to the revised paper on pp. 2.
>
> Q1.2.  At the beginning of Section 2.3 what is meant by "weak signals".
>
> >We are sorry for the confusion. During $l_{\infty}$-adversarial training, the model attempts to find the most robust feature against noise and drops the features which could be affected by adversarial noise. "Weak signals" means features that are non-robust to adversarial noise. We have revised this part in Section 2.3 on pp. 3.
>
> Q1.3. In Section 3.1 "In detail, we treat the input image..." seems to be 2 separate sentences? or the second half of the sentence "feature attribution methods aim to assign..." should be deleted?
>
> > Thanks for pointing out the typo! We have fixed it.
>
> Q1.4.In section 3.1, in the definition of regional inequality, what exactly does a region mean? Is it a set of pixels in the input space? How are these regions chosen?
>
> >The region is a block of input space that includes a set of pixels. The size of a region is set as 16*16 for experiments on ImageNet, and 4*4 for Cifar 10. We add the explanation of region in Section 3.1 on pp.4, and the setting of the region in experiments in Section 4.2 on pp.6,
>
> Q1.5. In 3.2.1, the text for the description of noise type 1 is confusing.
>
> > We are sorry for the confusion, and we have revised this part on pp.5
>
> **Q2.  Sparsity measure vs. Gini index & Motivation behind using Gini index.**
>
> >The sparsity of a vector can be quantified by $\frac{||x||_0}{|x|}$ [1], which simply calculates the ratio of non-zero elements.
> However, the sparsity measure treats an infinitesimally small value the same as a significant value. Even if some of the small coefficients increase by significant values, that change will not be reflected by a change in the value of the sparsity measure.
> For example, given a vector $x_1 = [0, 0, 0, 1, 1]$ and  $x_2=  [0, 0, 0, 1, 1000]$, the sparsity degree of $x_1$ and $x_2$ is equal to 0.4. However, their distributions are totally different. The distribution of $x_2$ is much more unequal compared with $x_1$.
> Also, $\frac{||x||_0}{|x|}$ is sensitive to noise, especially in settings where most values of elements are around 0 (e.g., feature attribution map).
>
> >In our case, Gini is able to reflect the change when a small coefficient increases. A Gini coefficient of 0 expresses perfect equality, where all values are the same, while a Gini coefficient of 1 (or 100%) expresses maximal inequality among values. For example, the Gini of $x_1 = [0, 0, 0, 1, 1]$ equals 0.6, and 0.799 for $x_2 = [0, 0, 0, 1, 1000]$. The value of Gini also provides an intuitive understanding of the distribution. When gini = 0.6, approximately 40% in the population (1-0.6 = 0.4) occupies the total worth. When gini = 0.799,  approximately 21.1% of the population dominates the worth. As for our experiment, gini of feature attributions by $l_{\infty}$-AT model ($\epsilon = 8.0$) is about 0.95, representing less than 5% of pixels that dominate the prediction.
>
> -----
> *[1] Duff, Iain S., Albert Maurice Erisman, and John Ker Reid. Direct methods for sparse matrices. Oxford University Press, 2017.*

---

> > ### Author Response · Authors · 2022-11-18
> > **Response to Reviewer 1FTk (Part II)**
> >
> > **Q3. Does this inequality phenomenon arise due to more sparse features or is a result of something more specific to Linf adversarial training. How successful are these occlusion and noise attacks on models trained with regularization for sparsity?**
> >
> > >A3: Thanks for your valuable question. Inspired by your question, we perform experiments on $l_{\infty}$-AT model and model trained with regularization for sparsity. We consider two types of sparse models: the model with sparse architecture and models with sparse weights. We compared $l_{\infty}$-AT model with models regularized by the following techniques:
> >
> > >- $l_1$ norm pruning [1]: It prunes filters by removing whole filters in the network together with their connecting feature maps.
> > >- Weight pruning [2]: It applies mask as regularization and sets the weights to be 0. It results in sparser weights and connectivity patterns.
> >
> > >| Model                | Clean acc | gini-g | gini-r |  Occ-B | Occ-W | Occ-B | Noise-I | Noise-II |
> > |----------------------|:---------:|:------:|:------:|:------:|:-----:|:-----:|:-------:|:--------:|
> > | $l_{\infty}$-AT model        | 54.53%    | 0.94   | 0.97   | 78.2%  | 80.2% | 60.2% | 60.9%   | 95.1%    |
> > | L1-norm pruning      | 73.08%    | 0.60   | 0.80   | 56.8%  | 57.8% | 52.5% | 40.7%   | 89.2%    |
> > | Weight-level pruning | 75.60%    | 0.62   | 0.81   | 33.1%  | 29.2% | 25.8% | 15.7%   | 49.9%    |
> >
> > >As shown in the Table, the sparsity of either model architecture or weights will not result in inequality on feature attribution. The $l_{\infty}$-AT model is much more easily affected by occlusion and noise attack than the two sparse models.
> > We think $l_{\infty}$ can be regarded as a strong regularization: during $l_{\infty}$-adversarial training, the model attempts to find the most robust feature against adversarial noise and discards the features which could be affected by added adversarial noise. With increasing the magnitude ($\epsilon$) of adversarial noise, only a handful of features $l_{\infty}$-adversarially trained model can rely on for recognition.
> >
> > **Q4.  It would be nice to see some visualizations for CIFAR-10 as well.**
> >
> > >A4: Thanks for your suggestion! We have added more visualization results for CIFAR-10 in Appendix A.3.
> >
> > **Q5. Could the authors provide clean and robust accuracies of models evaluated.**
> >
> > > A5: Sorry for the missing details! We added clean and robust accuracies of all the models mentioned in our work in Appendix A.1.
> >
> > **Q6. Filter out the images that both the clean model and adversarially trained model are incorrect on and then measure error.**
> >
> > > A6: Thanks for your suggestion! We did perform experiments as your suggestion for fair comparisons. We added this illustration in Sections 4.3 and 4.4 on pp. 7~8.
> >
> > -----
> > *[1] Hao Li, et al. "Pruning Filters for Efficient ConvNets."*
> >
> > *[2] Song Han, et al. "Learning Both weights and connections for Efficient Neural Network."*

---

> > > ### Author Response · Authors · 2022-11-18
> > > **Response to Reviewer 1FTk (Part III)**
> > >
> > > **Q7. Additionally, comparisons to L2 adversarially trained models can be interesting as well.**
> > >
> > > >A7: Thanks for your valuable suggestion.
> > > Due to different properties of $l_p$ normed vector space, the case of $l_2$ adversarial training is not the same as $l_{\infty}$ adversarial training. To be specific,  $l_{\infty}$ constrains the maximum magnitude of perturbation for each pixel. The adversarial noise is added on each pixel independently during the $l_{\infty}$ adversarial training. Therefore, the model attempts to find the most robust feature against noise and drops the features which could be affected by adversarial noise. Therefore, with increasing the magnitude ($\epsilon$) of adversarial noise, fewer but more robust features $l_{\infty}$-adversarially trained model can rely on for recognition.
> > >
> > > >Different from $l_{\infty}$ norm measures each pixel independently, $l_2$ norm calculates the square root of the inner product of all elements in a vector. Thus, during $l_2$ adversarial training, if a large budget of perturbation perturbs some pixels, the other pixels share the left budget on perturbation. [4] provides a game-theoretic understanding of $l_2$-adversarial training: during each loop of $l_2$-adversarial training, the attacker perturbs the features which are predictive for the model. When some predictive features are perturbed with the most budget of perturbation, the features perturbed with small or without perturbation are easier for the model to learn. Furthermore, these less perturbed features then become more predictive at the next training iteration. Thus, the inequality phenomenon does not occur during $l_2$-adversarial training. However, the $l_2$ adversarially trained model would use both the object-relevant and object-irrelevant features (e.g., background) for prediction at the game’s equilibrium. In [3] and [4], they show that $l_2$-adversarailly trained model is more sensitive to the background and other spurious features.
> > >
> > > >We add experiments about comparison with the $l_2$ adversarial trained model.
> > > >
> > > >| Model         | Clean acc | gini-g | gini-r |  Occ-B | Occ-G | Occ-W | Noise-I | Noise-II |
> > > |---------------|:---------:|:------:|:------:|:------:|:-----:|:-----:|:-------:|:--------:|
> > > | Std. trained  | 76.13%    | 0.62   | 0.84   | 33.0%  | 28.1% | 36.5% | 15.8%   | 49.8%    |
> > > | $l_2$-AT model   | 56.13%    | 0.60   | 0.76   | 51.4%  | 32.5% | 48.3% | 2.9%    | 36.3%    |
> > > | $l_{\infty}$-AT model | 54.53%    | 0.95   | 0.97   | 81.5%  | 61.0% | 77.2% | 75.6%   | 96.1%    |
> > > >
> > > >As explained above, the inequality degree of $l_2$-adversarial  trained model is similar to the standard trained model (or $l_2$ is even more equal). However, $l_2$-adversarial  trained model is still more vulnerable to occlusion. We think it is because the most influential features tend to cluster together for both  $l_{\infty}$ at and $ l_2 $ -adversarial model.
> > >
> > > ------
> > >
> > > *[3]. Moayeri, Mazda, et al "Explicit Tradeoffs between Adversarial and Natural Distributional Robustness."*
> > >
> > > *[4]. Moayeri, Mazda, et al. "A Comprehensive Study of Image Classification Model Sensitivity to Foregrounds, Backgrounds, and Visual Attributes."*

---

> > > > ### Comment · Reviewer_1FTk · 2022-11-18
> > > > **Thank you for the response**
> > > >
> > > > Thank you for the detailed response and for running additional experiments.  I greatly appreciate the efforts made by the authors in improving the clarity of the text.  In the camera-ready of the paper, could the authors also add the discussion of sparsity measure vs. Gini index in their response into the paper appendix and also add the discussion text for the sparsity regularization and l2 results into the appendix as well (currently the appendix only has the table results with no discussion)?  In the meantime, I am happy to increase my score to an 8 since all of my concerns were addressed.

---

> > > > > ### Author Response · Authors · 2022-11-19
> > > > > **Thanks a lot for the quick reply!**
> > > > >
> > > > > We are very happy and appreciate receiving your reply!
> > > > > We will add these discussions to the final version if the paper can be accepted successfully. Thanks a lot for recognizing our work! Have a good day!

---

### Official Review · Reviewer_Mxrb · 2022-10-24

**Confidence:** 4
**Correctness:** 4
**Technical Novelty And Significance:** 4
**Empirical Novelty And Significance:** 3
**Recommendation:** 8

**Clarity, Quality, Novelty And Reproducibility:**

The paper identified a novel phenomenon that do not exist in prior works. Description of the method is clear and easy to follow, the method can be reproduced.

**Strength And Weaknesses:**

Strength:
1. The strength of this paper is in the formulation of “inequality phenomenon” occurring during l_{infty} adversarial training. The paper provides an intuitive understanding about adversarial trained model’s behavior, and utilize a novel index to quantify such phenomenon.
2. Using Gini index is an extra strength, the author also extends original Gini index to regional and global aspects which makes technical contribution. The proposed index is strongly correlate with the inequality phenomenon.
3. Based on the observation of the phenomenon, the paper proposes several simple attack methods. The work identify unrealized threats brought by such inequality phenomena that $l_{\infty}$-adversarially trained models are much more vulnerable than standard trained models under inductive occlusion or noise. It shows unrealized vulnerability of adversarial training, which can inspire new research in this area.

Weakness:
1. Current attack methods create perceptible occlusions on resultant images, it could be good to see more imperceptible occlusions for performing attacks.
2. Writing can be improved at times. The paper could be further polished, and some typos exist.
3. More visualization results are preffered.


**Summary Of The Paper:**

Overall, the paper provides a novel and intuitive understanding about adversarial trained models. The paper identifies a new phenomenon and points out its unrealized threats which will inspire new related research.
Corresponding index (Gini coeffiencient) is designed to illustrate such phenomenon.


**Summary Of The Review:**

The paper undertakes an original approach to studying the inequality phenomenon of l_{infty} adversarial training and find new adversarial vulnerability. The phenomenon about adversarial training is novel, empirically insightful, and potentially will inspire further work. Overall, this new perspective is novel to me and I would tend to accept this paper. If the author can address my concerns. I would be more convinced.

---

> ### Author Response · Authors · 2022-11-18
> **Response to Reviewer Mxrb**
>
> **Q1: Current attack methods create perceptible occlusions on resultant images, it could be good to see more imperceptible occlusions for performing attacks.**
>
> >A1:Thanks for your valuable suggestion! On the one hand, our proposed occlusion attack is more stealthy in real-world scenarios.
> The imperceptible adversarial examples aim to create stealthy attacks in the real world, and the aim of our occlusion attack is similar to adversarial stickers [1]. Actually, our black/grey/white stickers are even more natural than crafted adversarial stickers with an obstructive pattern. On the other hand, this paper aims to study the threats caused by the inequality property of $l_{\infty}$ trained model. Therefore, optimizing the stealthiness of attacks is not our intention. We leave the improvement as future work, thanks!
>
> **Q2. Writing can be improved at times. The paper could be further polished, and some typos exist.**
>
>
> >A2: Thanks for your valuable comment, we have revised the paper and fixed all of the typos we found.
>
> **Q3. More visualization results are preferred.**
>
> >A3: Thanks for your valuable suggestion, we add more visualization results in Appendix A.2 and A.3.
>
> ------
> *[1]Eykholt, Kevin, et al. "Robust physical-world attacks on deep learning visual classification." Proceedings of the IEEE conference on computer vision and pattern recognition.  2018*

---

### Official Review · Reviewer_AXew · 2022-10-24

**Confidence:** 4
**Correctness:** 3
**Technical Novelty And Significance:** 3
**Empirical Novelty And Significance:** 3
**Recommendation:** 8

**Clarity, Quality, Novelty And Reproducibility:**

Clarity - Below average, due to grammatical errors
Quality and novelty - The finding in the paper is novel, although it is not clear whether these are fundamental differences that conflict with the robustness requirements.
Reproducibility - Clear

**Strength And Weaknesses:**

Strengths -

- The finding that Linf adversarial training makes models more vulnerable to noise and occlusion based attacks that perturb a few pixels, is novel and interesting.
- The proposed attacks show the difference between Linf-AT models and standard trained models.


Weaknesses -

- There are several grammatical errors that severely impact readability.
- Prior works have shown that gradients of adversarially trained models are perceptually aligned [1]. It is not clear why the feature attribution maps in Fig.1 are not perceptually aligned.
- If the perturbation radius is controlled such that only the non-robust features are suppressed during adversarial training, even though the model would be more vulnerable to occlusion and perturbation attacks when compared to standard training, it would actually be more aligned to human perception. For example, in a waterbirds vs. landbirds classification, a normally trained model may correctly classify an image even if important features of the bird are completely occluded, because it considers background and texture for classification. Whereas, an adversarially trained model may rely more on features such as beak, eyes and shape of the bird, making them more vulnerable to occlusion attacks. Despite this increased vulnerability, Linf-AT model is more aligned to human perception and hence is more preferred.
- There seems to be a typo in the paragraph above Eq.4. Sum of N least important features being similar to f(x) is considered as the criteria for inequality. This should have probably been the sum of N most important features I suppose?
- Could the authors show how the proposed attacks generated from an adversarially trained model perform on standard models, and vice versa?
- Could the authors confirm whether the standard trained models and adversarially trained models being compared in the paper use the same augmentations? This is important in light of the cutout augmentation results in A.3.
- It is important to show what happens in case of other adversarial training regimes such as L2 norm adversarial training.



[1] Tsipras et al., A. Robustness may be at odds with accuracy.


**Summary Of The Paper:**

This work highlights the limitations of L-inf adversarial training - models trained using Linf-AT (specifically at large perturbation bounds) tend to rely heavily on very few features, when compared to normally trained models. This makes them vulnerable to attacks that perturb very few pixels, or occlusion based attacks, which are indeed more common in the real world. The authors propose two attacks that replace important features with either noise or occlusions to show the vulnerability of Linf adversarially trained models.

**Summary Of The Review:**

Although the paper makes some novel and insightful observations, I believe it is not yet ready for publication in terms of clarity and completeness of experiments to draw conclusions - details are mentioned in the weaknesses section.

---

> ### Author Response · Authors · 2022-11-18
> **Response to Reviewer AXew (Part I)**
>
> **Q1: There are several grammatical errors that severely impact readability.**
>
> >A1: We regret there were problems with the English. The paper has been carefully revised to improve grammar and readability.
>
> **Q2. Visual conflict with results in [1].**
>
> >A2: Thanks for pointing out the confusion we ignored! Most visualization methods apply post-processing techniques during generating feature attribution maps. The post-processing technique is also clarified in [1]: “For CIFAR-10 and ImageNet, we clip gradients to within ±3σ and rescale them to lie in the [0, 1] range.” Thus, the most influential pixels with extremely high values are clipped to a relatively lower value (but they actually dominate the prediction). We compare feature attribution maps with and without post-processing introduced in Appendix A.2. As both Figures 5 and 6 show, the post-processed feature attribution maps are more perceptually-aligned with human observers.
>
> **Q3. Linf-AT model is more aligned to human perception and hence is more preferred.**
>
> >A3: We agree with your point, "The linf-AT model is more aligned to human perception". Compared to a standard-trained model, if we split each input image as object-relevant and object-irrelevant parts, $l_{\infty}$-adversarially trained model distinguishes the two parts better than the stndard trained model. However, though $l_{\infty}$-adversarially trained model is more aligned to human perception,  it is not as aligned with human perception as we expect.
>
> >As the reviewer expects: "an adversarially trained model may rely more on features such as beak, eyes and the shape of the bird".
> An ideal human perceptually aligned model is expected to make decisions based on a series of core feature attributions (beak, eyes, and the shape of the bird),  but not solely rely on individual one (only beak, or only eyes, or only shape of the bird). Although l_infty adversarial training leads the model to predict based on the object-relevant part, it actually utilizes very few features to make the prediction. For example, as Figure 4 shows, when we only occlude the beak of the bird or eye of the dog, $l_{\infty}$-AT model ($\epsilon = 8$) fails to recognize the images correctly.
> Our work reveals an interesting phenomenon that previous work ignored, which indicates a misalignment with human perception.
>
>
> **Q4 There seems to be a typo in the paragraph above Eq.4. Sum of N least important features being similar to f(x) is considered as the criteria for inequality. This should have probably been the sum of N most important features I suppose?**
>
> >A4: Thanks for pointing out the typo! We have fixed the typos in our revision.
>
> ------------------------------------------------------------------------------
>
> *[1] Tsipras et al., A. Robustness may be at odds with accuracy.*

---

> > ### Author Response · Authors · 2022-11-18
> > **Response to Reviewer AXew (Part II)**
> >
> > **Q5. Could the authors show how the proposed attacks generated from an adversarially trained model perform on standard models, and vice versa?**
> >
> > >A5: Thanks for your constructive suggestion! we add the following experiments: we perform occlusion attacks with two groups of attack budgets: Group 1: max cnt = 5,  r= 10, Group 2: max cnt = 10, r = 20. We perform noise attacks with threshold = 5000.
> > >
> > >| Attack | Occ-B | (cnt=5,r=10) | Occ-B  | (cnt=10,r=20) | Noise-I   |      | Noise-II  |       |
> > |--------|:------:|--------------|:------:|:-------------:|:-------:|:------:|:--------:|:------:|
> > | Model  |  Adv.  | Std.         |  Adv.  |      Std.     |   Adv.  |  Std.  |   Adv.   |  Std.  |
> > | Adv.(attacker)   | 100.0% | 11.4%        | 100.0% |     17.4%     |  100.0% |  10.6% |  100.0%  | 18.6%  |
> > | Std. (attacker)  |  22.8% | 100.0%       |  43.6% |     100.0%    |  26.8%  | 100.0% |   58.2%  | 100.0% |
> > >
> > >As the table shows, the transferability between the Linf-AT model and the standard-trained model is low. Transferring attack results from the standard-trained model to the Linf-AT model is easier. If the region of the most important pixels is occluded, the Linf-AT model fails to recognize the images correctly. The experiments are consistent with our observations.
> >
> > **Q6. Could the authors confirm whether the standard trained models and adversarially trained models being compared in the paper use the same augmentations?**
> >
> > >A6: We are sorry for the missing details about models, and we have added the details in Appendix A.1. The standard trained models and adversarially trained models being compared in the paper use the same augmentations during the training and testing phase.
> >
> > **Q7. It is important to show what happens in case of other adversarial training regimes such as L2 norm adversarial training.**
> >
> > >A7: Thanks for your valuable suggestion.
> > Due to different properties of $l_p$ normed vector space, the case of $l_2$ adversarial training is not the same as $l_{\infty}$ adversarial training. To be specific,  $l_{\infty}$ constrains the maximum magnitude of perturbation for each pixel. The adversarial noise is added on each pixel independently during the $l_{\infty}$ adversarial training. Therefore, the model attempts to find the most robust feature against noise and drops the features which could be affected by adversarial noise. Therefore, with increasing the magnitude ($\epsilon$) of adversarial noise, fewer but more robust features $l_{\infty}$-adversarially trained model can rely on for recognition.
> >
> > >Different from $l_{\infty}$ norm measures each pixel independently, $l_2$ norm calculates the square root of the inner product of all elements in a vector. Thus, during $l_2$ adversarial training, if a large budget of perturbation perturbs some pixels, the other pixels share the left budget on perturbation. [2] provides a game-theoretic understanding of $l_2$-adversarial training: during each loop of $l_2$-adversarial training, the attacker perturbs the features which are predictive for the model. When some predictive features are perturbed with the most budget of perturbation, the features perturbed with small or without perturbation are easier for the model to learn. Furthermore, these less perturbed features then become more predictive at the next training iteration. Thus, the inequality phenomenon does not occur during $l_2$-adversarial training. However, the $l_2$ adversarially trained model would use both the object-relevant and object-irrelevant features (e.g., background) for prediction at the game’s equilibrium. In [2] and [3], they show that $l_2$-adversarailly trained model is more sensitive to the background and other spurious features.
> >
> > >We add experiments about comparison with the $l_2$ adversarial trained model.
> > >
> > >| Model         | Clean acc | gini-g | gini-r |  Occ-B | Occ-G | Occ-W | Noise-I | Noise-II |
> > |---------------|:---------:|:------:|:------:|:------:|:-----:|:-----:|:-------:|:--------:|
> > | Std. trained  | 76.13%    | 0.62   | 0.84   | 33.0%  | 28.1% | 36.5% | 15.8%   | 49.8%    |
> > | $l_2$-AT model   | 56.13%    | 0.60   | 0.76   | 51.4%  | 32.5% | 48.3% | 2.9%    | 36.3%    |
> > | $l_{\infty}$-AT model | 54.53%    | 0.95   | 0.97   | 81.5%  | 61.0% | 77.2% | 75.6%   | 96.1%    |
> > >
> > >As explained above, the inequality degree of $l_2$-adversarial  trained model is similar to the standard trained model (or $l_2$ is even more equal). However, $l_2$-adversarial  trained model is still more vulnerable to occlusion. We think it is because the most influential features tend to cluster together for both  $l_{\infty}$ at and $ l_2 $ -adversarial model.
> >
> > ----------
> > [2]. Moayeri, Mazda, etl "Explicit Tradeoffs between Adversarial and Natural Distributional Robustness."
> >
> > [3]. Moayeri, Mazda, et al. "A Comprehensive Study of Image Classification Model Sensitivity to Foregrounds, Backgrounds, and Visual Attributes."

---

> > ### Author Response · Authors · 2022-11-25
> > **Additional Comments on Q3**
> >
> > We would like to add additional information about "whether $l_{\infty}$-AT model's feature attribution is perceptually aligned with humans".
> > We notice Gini itself is informative as we replied to Reviewer 1FTk:
> >
> > >The value of Gini provides an intuitive understanding of the distribution.
> > Gini coefficient of 0 expresses perfect equality, where all values are the same, while a Gini coefficient of 1 (or 100%) expresses maximal inequality among values.
> > For example, given a vector $x_1 = [0, 0, 0, 1, 1]$ and  $x_2=  [0, 0, 0, 1, 1000]$. The Gini of $x_1$ equals 0.6, and 0.799 for $x_2$. When gini = 0.6, approximately 40% of the population (1-0.6 = 0.4) occupies the total worth. When gini = 0.799, about 21.1% of the population dominates the worth. As for our experiment, gini of feature attributions by $l_{\infty}$-AT model ($\epsilon = 8.0$) is about 0.95, representing less than 5% of pixels that dominate the prediction. Namely, $l_{\infty}$-AT model relies on about one or two attributions (eye, nose) to make the prediction, which is different from human perception.
> >
> > Therefore, though $l_{\infty}$-AT  model uses the object itself to make the prediction, it actually relies on very few pixels, which is misaligned with human perception.

---

> ### Author Response · Authors · 2022-11-25
> **Whether our response addresses your concerns?**
>
> Dear reviewer AXew:
>
> We hope our response can address your concerns.
> Would you mind checking it and confirming if you have further questions?
> Please also let us know whether we have addressed your concerns at your convenience.
>
> Best Regards,
>
> Authors

---

> ### Comment · Reviewer_AXew · 2022-11-28
> **Post rebuttal update**
>
> I thank the authors for the detailed rebuttal which answers all my questions. I encourage the authors to include certain aspects such as attack transferability between standard and adversarially trained models, and comparison between L-inf and L2 training in the supplementary  section. It is interesting that despite the severe sparsity of features in adversarially trained models, they are still not a subset of features that are important for standard trained models (since attacks from ST models do not fool AT models).
>
> The authors have also made significant changes to improve the clarity and writing. However, there are several parts (such as the Global and Regional inequality bullets in page-4) which still have grammatical errors that impact the understanding of the reader. I encourage the authors to do a thorough review for the final version.
>
> I believe this work highlights an important weakness of L-inf adversarial training and would encourage more comprehensive training and evaluation methods in future. Therefore, I am happy to change my recommendation to "Accept".

---

> > ### Author Response · Authors · 2022-11-30
> > **Thanks for your response**
> >
> > Dear Reviewer AXew,
> >
> > We sincerely thank you again for your efforts in reviewing our paper and the valuable comments!
> >
> > We highly appreciate your careful consideration of the rebuttal. We are very glad to see that our answers addressed your concerns, and thank you for your feedback.
> >
> > Best Regards,
> >
> > Authors

---

### Official Review · Reviewer_yDNt · 2022-10-24

**Confidence:** 4
**Correctness:** 4
**Technical Novelty And Significance:** 3
**Empirical Novelty And Significance:** 3
**Recommendation:** 8

**Clarity, Quality, Novelty And Reproducibility:**

The paper introduces a novel phenomenon about l_{infty} adversarial training. This paper is well written and well organized, the method is easy to understand and reproduced.

**Strength And Weaknesses:**

Pros:
- A very interesting observation, inequality phenomenon occurring during l_{infty} adversarial training.
- Provides a new perspective on feature representation of l_{infty} adversarial trained model, which will motivate future research.
- Proposes regional and global Gini to evaluate the inequality phenomenon quantitively that providing an intuitive explanation.
- Experiments show l_{infty} adversarial trained model is even more fragile than standard trained model under some scenarios, the results are interesting for me.

Cons:
- Though this is a very relevant and timely work related to reliability of l_{infty} adversarial training, it would help if the authors could provide some effective solution to release such phenomenon or other suggestion.


**Summary Of The Paper:**

This paper provides some insights on the vulnerability of l_{infty} adversarial trained model. The paper identifies inequality phenomenon occurs during l_{infty} adversarial training which is quantified by Gini index. To show the later, the paper proposes two methods: inductive noise and occlusion to demonstrate the vulnerability of l_{infty} adversarial trained model caused by such phenomenon. This paper provides a novel perspective and sheds light on the practicality of l_{infty} adversarial trained model.

**Summary Of The Review:**

Interesting phenomenon, reasonable metric (Gini), well-motivated method that demonstrate the vulnerability of l_{infty} adversarial trained model.
This is a good paper, with some aspects of the presentation that should be improved.

---

> ### Author Response · Authors · 2022-11-18
> **Response to Reviewer yDNt**
>
> **Q1: It would help if the authors could provide some effective solution to release such a phenomenon or other suggestion.**
>
> A1: Thanks for your valuable question! In Sec. 5, we propose to combine Cutout during $l_{\infty}$-adversarial training to release such inequality. To be specific,  Cutout serves as a regularizer, and it forces the model learning features from different regions by cutting out part of training images at each iteration. The strategy slightly releases the inequality degree of the adversarially trained model. Moreover, we suggest $l_{\infty}$ adversarial training with a small adversarial attack budget $\epsilon$ could be preferred. Though less adversarial robust, $l_{\infty}$ adversarial training with a small $l_{\infty}$ introduces better human-aligned feature representation, and the inequality degree is moderate.
>
> **Q2: Some aspects of the presentation that should be improved.**
>
> A2: Thanks for your suggestion. We have revised the manuscript.

---

### Decision · Program_Chairs · 2023-01-20

**Decision:**

Accept: notable-top-25%

**Justification For Why Not Higher Score:**

Although the observation made regarding L_inf AT is interesting, its scope/contribution is a bit narrow for an oral.

**Justification For Why Not Lower Score:**

Reviewers gave high evaluations for this work.

**Metareview: Summary, Strengths And Weaknesses:**

The paper studies L_inf adversarial training and finds that few features dominate the prediction made by the adversarially trained model. This can be problematic in some cases. Reviewers generally liked the observation highlighting an important weakness of L-inf adversarial training. Authors did a good job in the rebuttal period and made significant changes to improve the clarity and writing. I think the paper could benefit by expanding the prior work section including other works showing some unexpected behavior of adversarially trained models. Given all I recommend an accept for this work.

**Note From Pc:**

if the above contains the word "oral" or "spotlight" please see: "oral" presentation means -> notable-top-5% and "spotlight" means -> notable-top-25%. As stated in our emails, we are disassociating presentation type from AC recommendations